# Shotgun Lipidomics for Differential Diagnosis of HPV-Associated Cervix Transformation

**DOI:** 10.3390/metabo12060503

**Published:** 2022-05-31

**Authors:** Natalia L. Starodubtseva, Vitaliy V. Chagovets, Maria E. Nekrasova, Niso M. Nazarova, Alisa O. Tokareva, Olga V. Bourmenskaya, Djamilja I. Attoeva, Eugenii N. Kukaev, Dmitriy Y. Trofimov, Vladimir E. Frankevich, Gennady T. Sukhikh

**Affiliations:** 1National Medical Research Center for Obstetrics Gynecology and Perinatology Named after Academician V.I., Kulakov of the Ministry of Healthcare of Russian Federation, 117997 Moscow, Russia; n_starodubtseva@oparina4.ru (N.L.S.); m_nekrasova@oparina4.ru (M.E.N.); n_nazarova@oparina4.ru (N.M.N.); alisa.tokareva@phystech.edu (A.O.T.); o_bourmenskaya@oparina4.ru (O.V.B.); d_attoeva@oparina4.ru (D.I.A.); e_kukaev@oparina4.ru (E.N.K.); d_trofimov@oparina4.ru (D.Y.T.); vfrankevich@gmail.com (V.E.F.); g_sukhikh@oparina4.ru (G.T.S.); 2Moscow Institute of Physics and Technology, 141700 Moscow, Russia; 3V.L. Talrose Institute for Energy Problems of Chemical Physics, Russia Academy of Sciences, 119991 Moscow, Russia; 4Department of Obstetrics, Gynecology, Perinatology and Reproductology, First Moscow State Medical University Named after I.M. Sechenov, 119991 Moscow, Russia

**Keywords:** lipidomics, human papillomavirus, cervical intraepithelial neoplasia, cervical cancer, diagnostics, mass spectrometry

## Abstract

A dramatic increase in cervical diseases associated with human papillomaviruses (HPV) in women of reproductive age has been observed over the past decades. An accurate differential diagnosis of the severity of cervical intraepithelial neoplasia and the choice of the optimal treatment requires the search for effective biomarkers with high diagnostic and prognostic value. The objective of this study was to introduce a method for rapid shotgun lipidomics to differentiate stages of HPV-associated cervix epithelium transformation. Tissue samples from 110 HPV-positive women with cervicitis (*n* = 30), low-grade squamous intraepithelial lesions (LSIL) (*n* = 30), high-grade squamous intraepithelial lesions (HSIL) (*n* = 30), and cervical cancers (*n* = 20) were obtained. The cervical epithelial tissue lipidome at different stages of cervix neoplastic transformation was studied by a shotgun label-free approach. It is based on electrospray ionization mass spectrometry (ESI-MS) data of a tissue extract. Lipidomic data were processed by the orthogonal projections to latent structures discriminant analysis (OPLS-DA) to build statistical models, differentiating stages of cervix transformation. Significant differences in the lipid profile between the lesion and surrounding tissues were revealed in chronic cervicitis, LSIL, HSIL, and cervical cancer. The lipids specific for HPV-induced cervical transformation mainly belong to glycerophospholipids: phosphatidylcholines, and phosphatidylethanolamines. The developed diagnostic OPLS-DA models were based on 23 marker lipids. More than 90% of these marker lipids positively correlated with the degree of cervix transformation. The algorithm was developed for the management of patients with HPV-associated diseases of the cervix, based on the panel of 23 lipids as a result. ESI-MS analysis of a lipid extract by direct injection through a loop, takes about 25 min (including preparation of the lipid extract), which is significantly less than the time required for the HPV test (several hours for hybrid capture and about an hour for PCR). This makes lipid mass spectrometric analysis a promising method for express diagnostics of HPV-associated neoplastic diseases of the cervix.

## 1. Introduction

A significant increase in cervical diseases caused by human papillomavirus (HPV) in women of reproductive age has been observed recently. HPV persistence often leads to malignancy and cervical cancer, which is one of the leading cancers and causes of death among women [1].

The cause of cervical cancer in 96% of cases is HPV of high carcinogenic risk [2]. According to the World Health Organization (WHO), the global prevalence of cervical intraepithelial neoplasia (CIN) is 40 million cases: CIN 1—30 million and CIN 2-3—10 million cases. The incidence of CIN 2–3 progression to carcinoma in situ varies from 40 to 60%. CINs accounts for 17 to 20% of all cervical pathologies in women of reproductive age. CIN 2–3 develops within 3 years in 27% of women after HPV infection [3].

Primary screening for precancerous and malignant diseases of the cervix in developed countries includes cytological examination of cervical smears and HPV testing [4]. Extended colposcopy is an in-depth method for cervical epithelium examination. The “gold standard” is the histological examination of the cervical biopsy. These methods have a number of significant disadvantages: subjectivity, difficulties in early diagnosis of the neoplastic process in high-risk groups, a lack of consensus on the management of patients at risk, the complexity of the differential diagnosis of CIN2 [5].

The cytological method is highly specific for the detection of severe lesions of the cervix uteri (high-grade squamous intraepithelial lesion, HSIL and squamous cell carcinoma, SCC) [6]. Replacing traditional cytology with liquid cytology has resulted in a reduction in inadequate smears and artifacts, an increase in SCC and CIN detection, and a decrease in false negatives [7]. Due to the low sensitivity of the cytological method, there is a danger of underestimating the severity of less pronounced anomalies of the cervical epithelium (atypical squamous cells of undetermined significance, ASCUS, and low-grade squamous intraepithelial lesions, LSILs) [8].

The results of long-term follow-up of 176,464 women aged 20–64 years from Sweden (Swedescreen), the Netherlands (POBASCAM), England (ARTISTIC), and Italy (NTCC) clearly showed that the most effective and progressive direction is the use of the HPV test in the primary manifestations of precancer and cervical cancer [9]. Compared with the cytological method of research, the HPV test has a higher sensitivity and reveals the main etiological factor in the development of precancer and cervical cancer. The disadvantage of this method is its low specificity.

The immunocytochemical (ICC) and histochemical (IHC) study of the expression of oncoproteins p16 and Ki-67 in scrapings and biopsy samples make it possible to determine the histogenesis of individual tumors and clarify the source of metastasis. This study takes a long time (on average, 7–14 days) and is quite costly. Thus, the search for effective and accurate minimally invasive biomarkers with high diagnostic and prognostic value, requiring minimal time and material costs, remains relevant.

More than 90% of malignant neoplasms result from genome damage—mutations. These are multiple mutations in the somatic cells of individual organs and tissues [10]. Research in the field of molecular genetics has found a specific range of genes, the mutations of which are related to malignant degeneration of cervical epithelium cells. The Cancer Genome Atlas (TCGA) project identified genes with a high level of mutations in cervical cancer (SHKBP1, ERBB3, CASP8, HLA-A, TGFBR2, PIK3CA, EP300, FBXW7, HLA-B, PTEN, NFE2L2, ARID1A, KRAS and MAPK1, SHBKBP1, HLA-A, and TGFBR2) [11]. Changes in the genome (somatic mutations and chromosomal rearrangements) lead to changes in the functional activity of genes, which can be estimated by the level of mRNA expression. We studied the activity of 13 genes involved in the processes of proliferation and regulation of the cell cycle (markers of proliferation Ki-67, MKI67; oncostatin M, OSM; cyclin-dependent kinase inhibitor 2A, CDKN2A,), apoptosis (B-cell lymphoma 2, BCL2; bcl-2-like protein 4, BAX; BAG family molecular chaperone regulator 1, BAG1), hormonal reception (progesterone receptor, PGR; estrogen receptor 1, ESR1), invasion (cathepsin V, CTSL2), inflammation and the immune response (prostaglandin-endoperoxide synthase 2, PTGS2; toll-like receptor 7 TLR7), and tumor suppression (phosphatase and tensin homolog, PTEN; signal peptide, SCUBE2).

Metabolomics (lipidomics) approaches have several advantages over proteomics and genomics presenting the distinct molecular phenotype of a particular pathological condition in real time [12]. The high diagnostic potential of tissue lipidomics has been shown in many areas of medicine [13,14,15], including oncology: lung cancer, thyroid gland, breast, stomach, pancreas, colorectal cancer, liver, kidney, prostate, ovarian cancer, endometrium, and HPV-associated head and neck cancer [16,17,18,19,20,21,22,23,24,25,26,27,28,29,30,31,32,33,34]. Cervical pre-cancer and cancer diagnostic potential was proven by multiple platforms including matrix-assisted laser desorption ionization (MALDI) [35], gas chromatography–mass spectrometry (GC–MS) [36], nuclear magnetic resonance (NMR) spectroscopy [37], and liquid chromatography-mass spectrometry (HPLC–MS) [13,38,39,40,41]. In particular, Porcari A.M. et al. developed a partial least squares discriminant analysis (PLS-DA) model for differentiating HSIL from normal cervical cytologic specimens, based on ceramides and sphingosine metabolites levels [13]. Methods used in these studies are laborious and time-consuming, limiting their application in cervical cancer screening. High diagnostic accuracy in cervical cancer tissue studies was demonstrated by ambient ionization MS: intelligent knife technique (iKnife) [42] and laser-assisted rapid evaporative ionisation mass spectrometry (LA-REIMS) [43]. Our preliminary studies found that the shotgun mass spectrometry (MS) lipid profile of cervical tissues differs significantly between tissues with benign processes (chronic cervicitis), precancerous diseases (LSIL and HSIL), and cancer [19,25]. The objective of this study was to introduce a method for rapid shotgun lipidomics to differentiate HPV-associated cervix epithelium transformation.

## 2. Results

### 2.1. Clinical Data

The average age (33.4 ± 7.0) did not differ statistically significantly in the groups studied: chronic cervicitis with HPV infection (ChC, *n* = 30), LSIL (*n* = 30), HSIL (*n* = 30), and squamous cervical cancer (SCC, *n* = 20). Comparative analysis of anthropometric data, menstrual function, hereditary, and obstetric anamnesis also did not reveal significant differences between the groups (*p*-value > 0.05) (Table 1). Vulvovaginal candidiasis was significantly more frequent in patients with HSIL compared with patients with LSIL (*p*-value = 0.03). Women with severe precancerous and malignant diseases of the cervix (HSIL and SCC) had a significantly greater number of sexual partners compared with chronic cervicitis and LSIL groups. HPV infection in history was found in more than half (53%) of the patients included in the study. Patients with HSIL had significantly more frequent HPV-infections in history, compared with the chronic cervicitis group (*p*-value = 0.03).

The results of the cytological study (Appendix A) are presented in Table 2. After extended colposcopy, all patients included in the study (*n* = 110) underwent a cervical biopsy (Appendix A). The specificity of the cytological method in patients with chronic cervicitis was 90%, in patients with LSIL it was also 90%, in patients with HSIL—81%, and in patients with SCC—99%. Despite the high specificity, the sensitivity of the cytological analysis was less than 75% for severe cervical transformation (HSIL and SCC) and less than 40% for chronic cervicitis and LSIL groups.

### 2.2. HPV Typing

The results of HPV typing are presented in Table 3 in accordance with the International Agency for Research on Cancer (IARC) classification (2012), which distinguishes HPV groups of high, as well as probable and possible carcinogenic risk. HPV of high carcinogenic risk (group 1) was detected for 85% of patients. More than 60% of HPV infections belonged to the A9 phylogenetic group (types 16, 31, 33, 35, 52, and 58). Most cases of HSIL (86%) and SCC (89.3%) were caused by a high carcinogenic risk HPV (group 1). The presence of several types of HPV simultaneously (mixed infection) was noted in 30% of patients. In the HSIL (*p*-value = 0.001) and SCC (*p*-value = 0.001) groups, the presence of two or more HPV types was significantly more common than in the ChC group.

The average HPV viral load was 5.3 Lg (1.6–8.8 Lg). More than half (53%) of the patients had a high viral load (more than 5 Lg): 34% of LSIL, 63% of HSIL, and 71% of SCC cases were due to long-term persistence of high-risk HPV with high viral load. HPV with a high viral load was significantly more common in the groups with severe dysplasia (HSIL) and SCC compared to the group with chronic cervicitis and LSIL (*p*-value < 0.05).

### 2.3. mRNA Expression during HPV Infection

Seven of the 13 genes studied had comparable expression levels in all groups. Statistically significant differences were obtained for the genes MKI67, CDKN2A, PGR, PTGS2, OSM, and PTEN (Figure 1). The level of CDKN2A mRNA expression was 4.7 times higher in HSIL (*p* = 0.014) and by 6.6 times (*p* = 0.001) higher in cervical cancer groups compared with chronic cervicitis. Pronounced changes were found in genes expression for cervical cancer compared to chronic cervicitis: a 7.3 times increase for MKI67 (*p*-value = 0.002), a 12 times decrease for PGR (*p*-value = 0.002), a 6.5 times decrease for OSM (*p*-value = 0.04), a 4 times decrease for PTGS2 (*p*-value = 0.02), and a 1.6 times decrease for PTEN (*p*-value = 0.001).

For 10 genes (CDKN2A, BCL2, PGR, TLR7, PTEN, OSM, PTGS2, ESR1, BAG1, and CTSL), a significant (*p*-value < 0.05) correlation with the degree of neoplastic transformation was found. The expression level of PGR, PTEN, and TLR7 changed most markedly with a histological diagnosis worsening.

An integral assessment of the expression profile of MKI67, PGR, CDKN2A, and BCL2 identified 45 patients (41%) at risk for developing cervical cancer: 14 (47%) from the LSIL group, 9 (30%) from the HSIL group, 22 (73%) patients from the cervical cancer group, and none of the patients from the group with cervicitis. At the same time, there was a statistically significant increase in the median risk index in the LSIL (*p* = 0.024), HSIL (*p* = 0.024), and cervical cancer (*p* = 8.8 × 10^−6^) groups compared with the cervicitis group (Figure 2).

### 2.4. Cervical Tissue Lipidomics

Characteristic positive ion mass spectra obtained by electrospray ionization mass spectrometry (ESI-MS) that averaged over 100 scans are shown in Appendix A. The most abundant peaks are observed in *m*/*z* 600–900 mass range. MS data were analyzed using orthogonal projections to latent structures discriminant analysis (OPLS-DA) for pairwise comparisons of pathological and surrounding tissues from patients of all considered groups. OPLS-DA analysis of the lipid profile of the lesion vs. the surrounding tissue revealed significant differences for each separate group (ChC, LSIL, HSIL, and SCC) (Appendix A).

The lipids with the maximum contribution to the OPLS-DA model division were identified for each group (Appendix A). Lipid species from five classes were observed. They are phosphatidylcholines (PC), lysophosphatidylcholines (LPC), phosphatidylethanolamines (PE), lysophosphatidylethanolamines (LPE), and sphingomyelins (SM). An increase in the level of all significant lipids was found in the LSIL dysplasia cervical tissue, the most pronounced increase was observed in PC and LPC (Appendix A). The peculiarities of the cervical epithelium lipid composition in severe neoplastic processes (HSIL) were a significant decrease in LPE, PC, and PE plasmalogens (PC–O and PE–O), and there were no differences in the levels of PC and sphingomyelins (SM) (Appendix A). The levels of PE and PE–O continued to decrease in comparison with the surrounding tissue, while the level of PC increased again in malignant tissue (SCC). (Appendix A).

OPLS-DA models were built to differentiate between benign (ChC), precancerous (LSIL and HSIL), and cancerous tissues (SCC) for the affected and surrounding tissues. The value of the dependent variable (“y”) in mathematical models was set in accordance with the severity of neoplastic transformation of the cervical epithelium (for ChC y = 0, for LSIL y = 1, for HSIL y = 2, and for SCC y = 3). In the case of surrounding tissues, the SCC samples were grouped separately from the ChC, LSIL, and HSIL sample cluster (Appendix A). Thus, the lipid composition of tissues adjacent to a malignant tumor undergoes significant changes.

A significant similarity between the HSIL and SCC lipidome was found for dysplasia cervical tissues (Appendix A). The levels of 23 lipids changed with an increase in the severity of cervical epithelium neoplastic transformation (VIP value > 1): phosphatidylcholines (PC 32:0, PC 32:1, PC 34:1, PC 34:2, PC 34:3, PC 34:4, PC 36:2, PC 36:3, PC 36:4, PC 36:5, PC 38:4, PC 38:5, PC 38:6, PC 40:5, and PC 40:6), ethanolamines (PE 38:4, PE 38:5, PE 38:6, and PE 40:6), lysophosphatidylcholines (LPC 16:0 and LPC 18:3), lysophosphatidylethanolamines (LPE 22:0), and plasmalogens (PC O-38:5) (Figure 3).

A significant positive correlation (r_s_ = 0.4; *p*-value < 0.001) was found between the MS prognosis calculated using the OPLS-DA model and the severity of intraepithelial lesions. Moreover, almost all marker lipids (85%) significantly (*p*-value < 0.05) negatively correlated with the degree of cervix transformation (Figure 4). In particular, Spearman’s coefficient for PC 36:5, PC 38:5, PCO-18:3, and LPC 18:3, the degree of epithelial lesion equal to r_s_ = −0.9, and for PE 38:6, PE 38:5, LPC 16:0, PC 38:5, PC 34:3, PC 40:6, PC 36:4, PC 34:2, PC 36:2, and PC 36:3—r_s_ = −0.6. Moreover, all these lipids demonstrated a strong significant positive correlation with each other. A certain synergy was found in the change in the lipid profile of the cervical epithelium during the transformation process triggered by HPV. Thus, ESI-MS demonstrates a high accuracy in the classification of neoplastic lesions of the cervix, comparable to the histological research method, which is the gold standard of diagnosis.

Based on the lipid profiles of the cervical tissues, an algorithm was developed for the differential diagnosis of HPV-associated diseases using OPLS-DA models: chronic cervicitis vs. neoplastic lesions; LSIL vs. HSIL and SCC; and HSIL vs. SCC (Figure 5). At the first stage, the presence of a benign (chronic cervicitis) or neoplastic process (LSIL, HSIL, and SCC) was determined using the appropriate OPLS-DA model for ESI-MS data. The sensitivity and specificity of this model were 97 and 87%, respectively. In the second step, an appropriate OPLS-DA model was used to classify mild neoplastic lesions of the cervix (LSIL) from severe precancerous and malignant diseases of the cervix (HSIL and SCC). The sensitivity and specificity of this model were 88% and 71%. The results of statistical validation of the OPLS-DA models by permutation analysis using 100 different model permutations show that the built models are valid.

## 3. Discussion

Infection with high-risk human papillomavirus is the main etiological cause of cervical cancer. The overexpression of viral proteins E6/E7 is believed to lead to oncogenesis. The E6 protein binds to p53 and the E7 protein interacts with the pRb protein, resulting in disruption of the balance between proliferation, regulation of the cell cycle and apoptosis, accumulation of somatic mutations, and chromosomal rearrangements [10,11].

In this study, we observed an increase in the mRNA expression of the MKI67 and CDKN2A genes with increasing severity of intraepithelial dysplasia, which is consistent with the results of other studies [44]. The combination of p16 and Ki67 detected by immunohistochemistry has been used in cytological samples to identify patients with high-grade cervical lesions [45].

Boon J.A et al. revealed the relationship between the severity of neoplasia and the level of CDKN2A, ESR, and PGR mRNA expression [46]. As the severity increased, CDKN2A expression increased and ESR and PGR expression decreased. Our results are consistent with the data of the Boon J.A et al. study for the CDKN2A and PGR genes.

A decrease in the level of PTEN mRNA expression during the development of HPV-associated transformation of the cervical epithelium was expected. Phosphatase and tensin homolog (PTEN) negatively regulate intracellular levels of phosphatidylinositol-3,4,5-trisphosphate in cells and functions as a tumor suppressor by negatively regulating the AKT/PKB signaling pathway. It is also one of the most frequently mutated genes in human cancer. PTEN regulates many cell processes: growth, apoptosis, migration, adhesion, and invasion [47,48,49]. Li-na Peng et al.l observed that PTEN was significantly downregulated in the cervical carcinoma tissues [50].

Oncostatin-M (OSM) utilizes JAK–STAT3 and PI3K–AKT–mTOR pathways to promote EMT-associated cancer cell invasion and metastasis [51]. One of the most common genomic imbalances in cervical SCC is copy number gain and the amplification of chromosome 5p, which occurs in up to half of advanced-stage cervical SCCs. The OSMR gene is located on the short arm of chromosome 5 (region 5p13.1). Overexpression of the oncostatin-M receptor (OSMR) in SCC cells results in an increased sensitivity to the major ligand OSM [52,53]. It is possible that our study included patients without metastases and with a more favorable prognosis; therefore, a high level of OSM expression was not found.

Cyclooxygenase COX-2 (also known as PTGS2) is an enzyme that catalyzes the conversion of arachidonic acid to prostaglandins and modulates several cellular processes: cell cycle regulation, apoptosis, extracellular matrix deposition, and angiogenesis. Prostaglandins have an important role in cancer-related progression and inflammation. PTGS2 over-expression has been considered as an indicator of invasiveness, aggressiveness, and metastatic potential in different malignancies including cervical carcinoma [54,55]. In our study, we observed a statistically significant decrease in PTGS2 expression. This could be due to two reasons: the comparison with chronic cervicitis and the fact that we examined samples of SCC of the cervix. PTGS2 is more frequently expressed in adenocarcinomas than in SCCs (57–94% vs. 24%) [55,56]. We have previously shown a gradual increase in the expression of proliferation markers and a decrease in the expression of proapoptotic genes (estrogen receptor ESR1, progesterone receptor PGR, PTEN, and PTGS2) for progressive degrees of cervical intraepithelial neoplasia leading to cancer [57]. A model was proposed for assessing the risk of pathology progression utilizing the level of mRNA expression for MKI67, CDKN2A, PGR, and BAX genes.

The lipidomic analysis of cervical tissues neoplastic changes is considered as a highly informative method for biomarker search [19,25]. Lipids in the cell are the main constituent of the membrane and are responsible for energy storage and signaling during cell growth, inflammation, and the immune response [58]. Increased lipid biosynthesis is a specific feature of cancer [59,60,61,62,63]. Increased synthesis of fatty acids is necessary for the rapid proliferation of tumor cells, providing them with a substrate (in particular, phospholipids) for membranes [59,60,64].

A significant increase in the level of phosphatidylcholines was found in cervical cancerous tissues. T. Altadill and colleagues obtained similar results for endometrial tumor tissues. The possibility of differentiating the stages of endometrial cancer (according to FIGO) by the lipid profile of endometrial tissue samples has been demonstrated also [33].

The increase in the PC 34:1 level during cervical epithelium neoplastic transformation was consistent with data from other studies. In particular, an increase in the level of this lipid in cancerous tissues of the breast, endometrium, and thyroid gland was noted [17,22,23,65]. PC 32:0 was elevated in cancerous tissues of the cervix. Similar results were presented by E. Cífková, J. Ryu, and L. Krasny in the study of breast, thyroid, and HPV-associated head and neck cancer lipidomics [66,67,68]. The level of LPC 16:0 in the cancerous epithelium was reduced. This result is consistent with the data of T. Goto and Y. Morita, obtained for cancerous tissues of the prostate gland and liver [32,69].

In our study, the main changes in malignant transformation of the cervical epithelium affected glycerophospholipids. This class of lipids has previously been proposed as potential tumor markers for other types of cancer (breast, thyroid, prostate, ovarian, endometrial, liver, pancreas, kidney, squamous cell lung cancer, stomach cancer, colorectal cancer, and HPV-associated squamous cell head cancer and neck) [16,17,18,19,20,21,22,23,24,25,26,27,28,29,30,31,32,33,34].

L. Krasny et al. studied tissue lipidomics in cancer and benign diseases of the parotid gland [67]. Parotid cancer is one of the so-called head and neck cancers. HPV plays a key role in the pathogenesis of head and neck tumors, since about 72% of cases are associated with hr-HPV [70]. Mass spectrometric analysis of parotid cancer tissues revealed 243 significantly altered lipid species. Most of them were glycerophospholipids, followed by glycerolipids and sphingolipids. The results of this and our research showed that the revealed changes in lipid levels according to MS analysis reflect morphological changes in cancer tissues. [66].

## 4. Materials and Methods

### 4.1. Chemicals and Reagents

Acetonitrile, 2-propanol, chloroform, methanol, water, and NaCl were of high purity grade and purchased from Sigma–Aldrich (Steinheim, Germany). PREP-NA, Proba-GS, PCR HPV “Kvant-21” kits were purchased from DNA-TECHNOLOGY LLC, Russia.

### 4.2. Study Design

All patients (*n* = 110) read and signed voluntary informed consent. The study was approved by the Ethical Committee of the National Medical Research Center for Obstetrics, Gynecology and Perinatology named after Academician V.I. Kulakov (protocol No. 4 of October 2017). All patients were HPV-positive. Four groups were formed depending on the results of histological examination of cervical biopsies: chronic cervicitis with HPV infection (ChC, *n* = 30), LSIL (*n* = 30), HSIL (*n* = 30), and squamous cervical cancer (SCC, *n* = 20).

The inclusion criteria were the following: reproductive age (21–45 years); HPV infection; intraepithelial lesions of the cervix, confirmed by a histological study; regular menstrual cycle; ability to comply with the protocol requirements; and written informed consent. Pregnancy, lactation, postpartum period, hormonal therapy, acute inflammatory diseases of the pelvic organs, impaired renal, liver, lung function in the stage of decompensation, and neuropsychiatric diseases were exclusion criteria.

### 4.3. Morphological Investigation

Evaluation of cytological smears (Cervix-brush type) from the cervix was carried out according to the Terminology Bethesda System (TBS) proposed by the US National Cancer Research Institute (last revised in 2014). Cytological smears were divided into three categories: normal (NILM), smears of undetermined value (ASCUS—atypical cells of undetermined significance), intraepithelial lesions (precancerous), low (LSIL—low-grade squamous intraepitelial lesions), and high (HSIL—high-grade squamous intraepitelial lesions) degree.

Cervical biopsies were scored as follows: mild dysplasia (CIN1), moderate dysplasia (CIN2), severe dysplasia (CIN3), and cervical cancer. Biopsy material was assessed using Lower Anogenital Squamous Terminology (LAST) (2012), where CIN1 is equivalent to LSIL and CIN2 and CIN3 are equivalent to HSIL [71].

### 4.4. HPV Typing

The Proba GS kit (DNA-Technology, Russia) was used to isolate DNA. Cells were lysed with a strong chaotropic agent, nucleic acids from the cells were adsorbed on a sorbent, washed, and eluted. The volume of samples after isolation was 100 μL.

Amplification of type-specific DNA fragments of human papillomavirus and human DNA used to control sampling was carried out using the Kvant-21 reagent kit (DNA-Technology, Russia) for the detection, typing, and quantitation of HPV by PCR. Amplification was performed in real time using a DT-964 instrument (DNA-Technology, Russia). The fluorescence level was measured at each cycle of amplification in the FAM, HEX, ROX, and Cy5 channels. The processing of the results was carried out automatically using the software for the DT-964.

### 4.5. mRNA Expression Analysis

The level of mRNA expression of 13 genes (CDKN2A, MKI67, BCL2, PGR, TLR7, PTEN, OSM, PTGS2, SCUBE2, BAX, ESR1, BAG1, and CTSL2) in scrapings of the cervical epithelium was determined by the reverse transcription and polymerase chain reaction (RT-PCR). The material was taken with a cytobrush from the cervix, with the obligatory capture of the transformation zone (the junction of two epitheliums—multilayer flat and cylindrical). To prevent mRNA degradation, the brush was immediately immersed in a test tube with guanidine thiocyanate solution (lysis solution, Proba-NK kit, DNA-TECHNOLOGY LLC, Moscow, Russia), rotated in a test tube for 10–15 s, and removed by squeezing thoroughly. The samples were stored at −20 °C for one month or at −70 °C for one year.

RNA isolation was performed using the PREP-NA kit (DNA-Technology, Moscow, Russia). The method is based on lysis in a guanidine thiocyanate solution and precipitation of nucleic acids. Reverse transcription was performed at 40 °C for 30 min and then at 95 °C for 5 min. PCR was carried out by thermocycler RealTime system DTprime 4X1 in the “real time” mode in a volume of 12 μL according to the program: 15 cycles-80 °C 5 s, 94 °C 5 s; 1 cycle-94 °C 5 min; 50 cycles-94 °C 20 s, 64 °C 20 s; 10 °C-storage. The following reagents from DNA-Technology, Russia were used: RT-buffer, PCR-buffer, dNTPs, reverse transcriptase, Taq-polymerase, oligonucleotide RT primers and PCR primers, and fluorescent-labeled TaqMan-probes corresponding to each gene (CDKN2A, MKI67, BCL2, PGR, TLR7, PTEN, OSM, PTGS2, SCUBE2, BAX, ESR1, BAG1, and CTSL2). The fluorescence level was measured on each cycle at a temperature of 64 °C along the FAM fluorescence channel. To increase the sensitivity and specificity of PCR, a “hot start” was used, which was provided by polymerase with antibodies (TechnoTaq).

### 4.6. Tissue Preparation for Lipidome Analysis

Part of the punch biopsy specimen for histological examination (10 mg) was frozen in liquid nitrogen and stored at −80 °C for further lipidomic research. Lipid extracts were obtained in accordance with the modified Folch method [72]. A biopsy material of about 10 mg was homogenized in a ceramic mortar with liquid nitrogen, following the addition of 4 mL of a chloroform–methanol mixture (2:1, *v*/*v*), incubation for 10 min, and filtration using filter paper. A volume of 800 μL of an aqueous solution of NaCl (1 mol/L) was added to the filtrate to induce phase separation. The mixture was centrifuged at 400× *g* for 5 min at room temperature. The lower organic layer containing lipids was collected and dried in a stream of nitrogen, then redissolved in 40 μL of acetonitrile-2-propanol (1:1, *v*/*v*) for mass spectrometric analysis. The samples of tissue were very small (punch biopsy) in order to minimize trauma to the cervix, especially with small lesions of the epithelium. Thus, the final sample volume was 40 µL and was only sufficient for analysis in one ion mode.

### 4.7. Mass Spectrometric Analysis of Lipid Extracts

Molecular composition of the samples was determined by flow injection analysis (FIA) electrospray ionization mass spectrometry using Maxis Impact qTOF mass spectrometer (Bruker Daltonics, Bremen, Germany) [22,23,24,25]. Constant flow of methanol/water 9/1 was supplied with rate of 10 µL/min by Dionex UltiMate 3000 binary pump and 20 µL of sample was injected by Dionex UltiMate 3000 autosampler (ThermoScientific, Bremen, Germany). Mass spectra were obtained in the positive ion mode over the mass range *m*/*z* 400–1000 with resolution of 50,000 and the following ion source settings: capillary voltage 4.1 kV, spray gas pressure 0.7 bar, drying gas flow rate 6 L/min, and the temperature of the drying gas was 200 °C. Due to the low sample volume the positive ion mode was chosen for this study only.

Tandem mass spectrometry was done using data-dependent analysis with the following characteristics. Five of the most abundant peaks were chosen after full mass scan and subjected to MS/MS analysis with collision induced dissociation applying 35 eV collision energy, 1 Da isolation window, and 1 min of mass exclusion time.

After the MS analysis, 100 mass spectra obtained during a sample elution were averaged, normalized by total ion current (TIC), and transformed into the abundance-*m*/*z* table for further processing (Appendix A). The spray was stable throughout the analysis as the flow was constant and the ion source parameters were not changed. The relative standard deviation of TIC over the integration time did not exceed 10%. The intragroup relative standard deviation was about 10–20%.

The lipids were annotated with in-lab created R code (the RStudio version was 1.1.463 and the R language version was 3.5.2), which compares measured accurate *m*/*z* values with theoretical computer-generated values. The code searched a record within 10 ppm from the experimental *m*/*z*. Proton and sodium ion adducts were considered. More precise identification was done based on the MS/MS data for the peak under consideration, if it had undergone MS/MS analysis. Lipids nomenclature through the paper is in accordance with LIPID MAPS [73] terminology and shorthand notation summarized in [74]. Annotated lipids were used for further statistical analysis.

### 4.8. Statistical Analysis

Descriptive statistics are presented as mean values (M) and their standard deviations (δ). Frequency rates (%) were determined for qualitative data. The median (Me) was used as a measure of the central tendency of all quantitative indicators, and the lower Q1 (0.25) and upper Q3 (0.75) quartiles were used as an interval estimate.

The Mann–Whitney U-test was used to assess the reliability of intergroup differences in quantitative indicators. Differences were considered statistically significant at *p*-value < 0.05. To assess the reliability of intergroup differences in qualitative indicators, the Chi-square test was used, adjusted for continuity.

The quantitative assessment of the mRNA expression was carried out using the ΔCq method with normalization for the three reference genes TBP, B2M, and GUSB. Median in the cervicitis group, which was calculated according to the 2^−ΔΔCT^ method. The median value in the cervicitis group was equated to 1. An integral assessment of the expression level of four genes (MKI67, CDKN2A, PGR, and BCL2) was carried out to assess the risk index (RI) for the development of neoplastic transformation of the cervical epithelium [57] according to the formula:RI = 0.8 ∗ ln [MKI67]/[PGR] + 1.6 ∗ ln [CDKN2A]/[BCL2] − 4(1)

[MKI67]/[PGR]—a ratio of mRNA expression levels MKI67 and PGR,

[CDKN2A]/[BCL2]—a ratio of mRNA expression levels of CDKN2A and BCL2.

The IR threshold was previously calculated using ROC analysis. It is equal to 57 units on a 100-point scale.

ESI-MS data were analyzed using multivariate latent structures discriminant analysis (OPLS-DA) for pairwise comparisons of pathological and surrounding tissue categories and to compare surrounding tissues and pathological tissues with each of the diagnoses (ChC, LSIL, HSIL, and SCC) [25,75]. These methods allow to make a statistical model to classify the studied samples. The ions with variable importance in projection (VIP) higher than 1 were considered as significant for classification annotated and used for further analysis. The OPLS-DA model performance is assessed by its ability to fit (R^2^) and predict (Q^2^) variance of the data. The Q^2^ parameter was calculated by 7-fold cross-validation. Permutation test with 100 permutations was used to validate models and estimate the significance of Q^2^ and R^2^.

The degree of correlation of potential marker lipids for a neoplastic process with histological diagnosis and expression of 13 studied genes was assessed using Spearman’s rank correlation [76]. Lipids with significant correlation (*p*-value < 0.05) were used to construct OPLS-DA models for categorical prediction of the diagnosis [25]. To determine the sensitivity and specificity of the obtained models, 10 cross-validation procedures were carried out (samples were divided in a ratio of 1:9 in the training/validation samples) with subsequent averaging of the results. Statistical data processing was performed using Microsoft Excel tables and Statistica for Windows v.7.0 software packages, StatSoft Inc. (Tulsa, OK, USA), IBM SPSS v.22.0, in the R.

## 5. Conclusions

Shotgun lipidomics were proven to have a high potential for accurate, fast, and minimally invasive early and differential diagnosis of HPV-associated cervical diseases. Significant differences in the lipid profile between the affected and surrounding tissues were revealed in chronic cervicitis, LSIL, HSIL, and cervical cancer. ESI-MS analysis of lipid extract by direct injection takes about 25 min (including preparation of the lipid extract), which is significantly less than the time required for the HPV test (several hours for hybrid capture and about an hour for PCR). This makes lipid mass spectrometric analysis a promising method for the express diagnostics of HPV-associated neoplastic diseases of the cervix.

In this study, the lipids specific for HPV-induced cervical transformation mainly belong to glycerophospholipids: PC and PE. These classes of lipids are associated with apoptosis suppression, impaired cell metabolism, and the stimulation of proliferative processes. An increase in the level of PC and a decrease in the level of LPC in tissues with neoplastic transformation, were shown for tumor tissues of other organs.

The lipid composition of tissues adjacent to the transformation zone undergoes significant changes. At the same time, the lipid spectra of the affected tissues in severe lesions (HSIL) and cervical cancer overlapped in many ways. Diagnostic OPLS-DA models were developed based on 23 marker lipids. More than 90% of these marker lipids positively correlated (*p*-value < 0.05) with the degree of cervix transformation. Thus, the ESI-MS of changed tissue organic extract demonstrates high accuracy in the classification of neoplastic lesions of the cervix, comparable to the histological research method, the gold standard of diagnosis.

As a result, an algorithm was developed for the management of patients with HPV-associated diseases of the cervix, based on the panel of 23 lipids. Lipidomic analysis of the affected cervical tissue by ESI-MS, along with HPV-typing and histological examination, can be of great practical importance for early, differential, accurate, and rapid diagnosis of the severity of cervical epithelium dysplasia. The results of the work will contribute to the early detection of HSIL and cervical cancer, a more accurate diagnosis (LSIL vs. HSIL), as well as a decrease in the unnecessary use of destructive treatment techniques.

## Figures and Tables

**Figure 1 metabolites-12-00503-f001:**
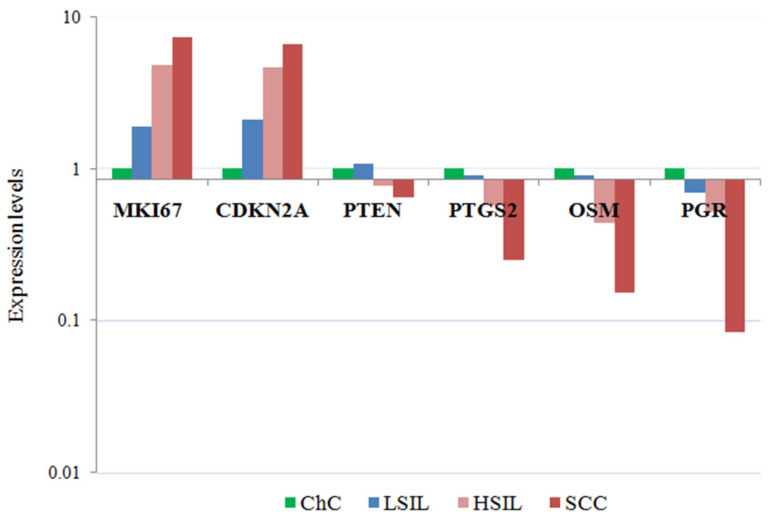
mRNA expression levels in cells of cervical smears with lesions of the cervical epithelium. The medians of the level of gene expression with statistically significant differences in the study groups are presented.

**Figure 2 metabolites-12-00503-f002:**
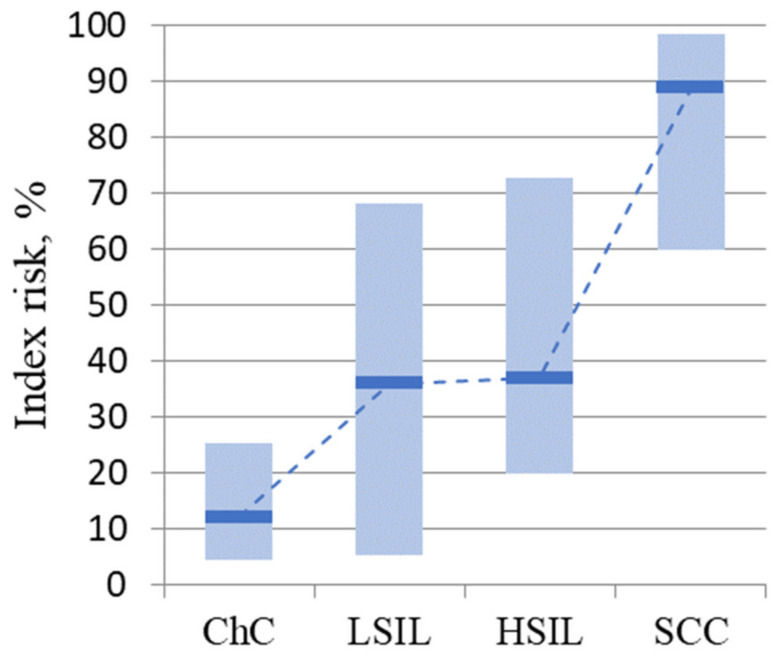
Risk index values for disease progression by group. The medians in the study groups and the interquartile range are presented.

**Figure 3 metabolites-12-00503-f003:**
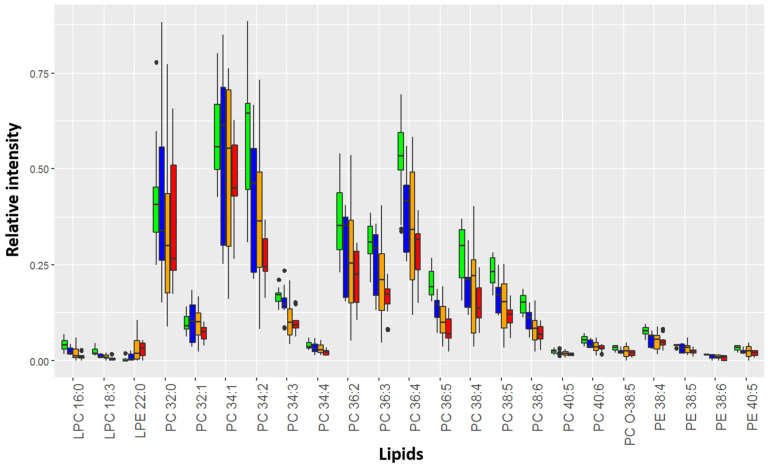
Relative intensity of marker lipids in the mass spectrum of affected tissues. Green corresponds to chronic cervicitis, blue to LSIL, yellow to HSIL, and red to SCC. The diagram shows Q1–1.5*IQR, Q1, Me, Q3, and Q3 + 1.5*IQR. Black dots correspond to outliers.

**Figure 4 metabolites-12-00503-f004:**
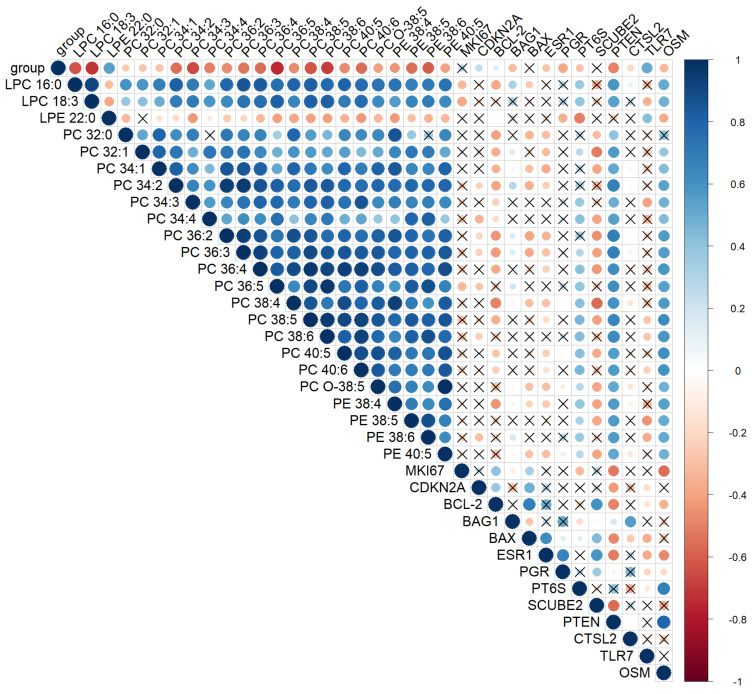
Spearman’s correlation analysis of the cervical tissues’ lipid profile in CINs and SCC, histological diagnosis, and gene mRNA expression levels at a confidence level of 0.05. Positive correlation is highlighted in blue and negative correlation in red. The degree of correlation is highlighted in color—the stronger the correlation, the darker the color. “X”—correlations with *p*-value > 0.05. Parameter group shows the degree of cervical epithelium transformation: 0-ChC, 1-LSIL, 2-HSIL, and 3-CC.

**Figure 5 metabolites-12-00503-f005:**
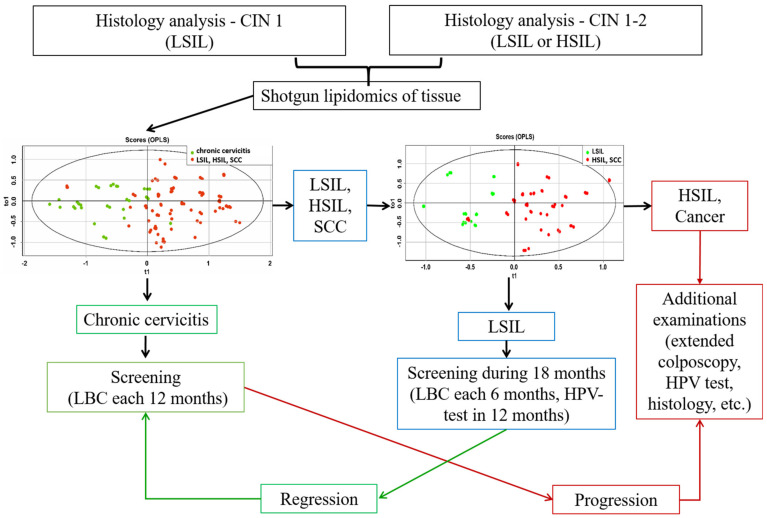
Algorithm for management of patients with HPV-associated cervix transformation including the OPLS-DA models, based on ESI-MS data of cervical tissue extracts.

**Table 1 metabolites-12-00503-t001:** Demographic and clinical data of the patients. There were no significant differences between the groups (*p*-value > 0.05).

Parameter	ChC (*n* = 30)	LSIL (*n* = 30)	HSIL (*n* = 30)	SCC (*n* = 20)
Age, years	29 ± 3.7	32 ± 4.8	34 ± 3.2	37 ± 3.3
Height, cm	167.6 ± 3.9	167.2 ± 3.8	167.4 ± 4.5	166.4 ± 5.8
Body mass, kg	62.3 ± 7.0	63.1 ± 10.5	63.3 ± 11.6	63.6 ± 11.2
Menarche, years	12.9 ± 1.0	13.1 ± 1.3	12.8 ± 1.1	13.2 ± 0.9
Menstrual cycle length, days	29.2 ± 2.4	28.4 ± 1.8	28.7 ± 2.1	27.7 ± 3.5
Duration of menstruation, days	5.2 ± 0.8	5.4 ± 1.1	5.3 ± 0.9	5.4 ± 0.5
Number of pregnancies	45 (21%)	39 (18%)	56 (27%)	73 (34%)
Number of spontaneous births	23 (23%)	23 (23%)	25 (25%)	30 (29%)
Number of induced abortions	9 (12%)	7 (9%)	28 (36%)	33 (43%)

**Table 2 metabolites-12-00503-t002:** Results of cytological examination.

Cytological Examination	ChC, *n* = 30	LSIL, *n* = 30	HSIL, *n* = 30	SCC, *n* = 20
NILM	6 (20%)	3 (10%)	1 (3.3%)	1 (5%)
Chronic cervicitis	11 (37%)	6 (20%)	1 (3.3%)	1 (5%)
ASCUS	7 (23%)	6 (20%)	2 (7%)	-
LSIL	4 (13.4%)	12 (40%)	4 (13.4%)	-
HSIL	1 (3.3%)	3 (10%)	22 (73%)	3 (15%)
SCC	1 (3.3%)	-	-	15 (75%)

**Table 3 metabolites-12-00503-t003:** Results of HPV testing in study groups according to the IARC carcinogenicity classification.

HPV Groups for for Carcinogenicity	HPV Phylogenetic Group	HPV Type	ChC, *n* = 30	LSIL, *n* = 30	HSIL, *n* = 30	SCC, *n* = 20	Total, *n* = 110
1	A9	16	5 (16.7%)	9 (30%)	21 (70%)	11 (55%)	46 (42%)
		52	2 (6.7%)	4 (13.4%)	-	-	6 (5.4%)
		33	-	1 (3.3%)	5 (16.7%)	2 (10%)	8 (7.3%)
		58	2 (6.7%)	4 (13.4%)	1 (3.3%)	-	7 (6.4%)
		31	2 (6.7%)	3 (10%)	2 (7%)	2 (10%)	9 (8.2%)
		35	2 (6.7%)	1 (3.3%)	4 (13.4%)	2 (10%)	9 (8.2%)
2A	A7	68	1 (3.3%)	1 (3.3%)	-	1 (5%)	3 (2.7%)
1	A7	45	2 (6.7%)	-	2 (7%)	1 (5%)	5 (4.5%)
		18	3 (10%)	2 (7%)	-	4 (20%)	9 (8.2%)
		59	-	1 (3.3%)	1 (3.3%)	-	2 (1.8%)
		39	2 (6.7%)	-	-	-	2 (1.8%)
2B	A6	66	-	-	2 (7%)	1 (5%)	3 (2.7%)
1		56	4 (13.3%)	3 (10%)	3 (10%)	-	10 (9%)
2B		53	1 (3.3%)	1 (3.3%)	-	1 (5%)	3 (2.7%)
LR	A10	6	-	-	-	1 (5%)	1 (0.9%)
LR		44 (55)	1 (3.3%)	3 (10%)	2 (7%)	2 (10%)	8 (7.3%)
2B	A5	82	-	2 (7%)	1 (3.3%)	-	3 (2.7%)
1		51	3 (10%)	-	5 (16.7%)	-	8 (7.3%)

## Data Availability

Data is contained within the Appendix A.

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
