# Peer review of "Shotgun Lipidomics for Differential Diagnosis of HPV-Associated Cervix Transformation"

_metabolites, 2022, doi:10.3390/metabo12060503_

Round 1

Reviewer 1 Report

The authors improved the manuscript significantly since the last version and answered all my previous questions. It seems that the models can work and be applied in medical diagnostics.

Author Response

Thank you very much for your valuable remarks and suggestions

Reviewer 2 Report

The manuscript entitled "Shortgun lipidomics for differential diagnosis of HPV-associated cervix transformation" describes a lipidomic approach to HPV diagnosis.

The manuscript is well written and organized, the results are also fundamented with the reported in literature.

In this sense, in my opinion it should be accepted after minor revisions.

Comments:

  • All abbreviation shoulb be described after when used for the first time
  • Figure 1 is too big and the font number is different from the main text. Please correct, 
  • The same for figure 2; there is no units for y axis?
  • In english the numbers are divided by points not comma. P*lease correct

Author Response

Comments:

  1. All abbreviation should be described after when used for the first time

Answer: abbreviations were explained at their first appearance.

  1. Figure 1 is too big and the font number is different from the main text. Please correct.

Answer: corrected.

  1. The same for figure 2; there is no units for y axis?

Answer: corrected, units for y axis were added.

  1. In english the numbers are divided by points not comma. Please correct

Answer: corrected.

Reviewer 3 Report

The manuscript “Shotgun lipidomics for differential diagnosis of HPV-associated cervix transformation” provided by Natalia L. Starodubtseva et al. provides a promising mass spectrometry based method for diagnostics of HPV-associated neoplastic diseases of the cervix. The literature characterizes 110 HPV-positive clinical samples and found lipid biomarkers for with HPV-associated diseases of the cervix.

In introduction section, the author started from the importance of HPV testing, and the advantage and disadvantage of the current testing methods. Also, the genes from previous studies related to cervical 39 cancer were included. Moreover, the author summarized the application of mass spectrometry based metabolomics (lipidomics) reported in cervical cancer study and their limitations. The result section clearly showed the characterization of the clinical samples, and significant mRNA expression among different kind of tissues. Mass spectrometry method identified 23 lipid biomarkers for affected tissue and the correlation showed the trustworthy of the results. The author proposed an algorithm for the differential diagnosis of HPV-associated diseases based on this study. Most of the result description were supported by the data/figure. In discussion part, the author cited previous reported results to support the conclusions made from the current study. Method section provides a clear and precise description of the assay process in the study.

This manuscript is overall well-written, and I have a few questions and suggestions to make as following:

In Figure 1, is the x-axis label “0,1” indicates “0.1”? “12 times decrease for PGR” in the result description is about 0.083. If the lowest value in x-axis is 0.1, 0.083 is out of range.

In supplemental table 1, all the lipids detected were indicated as “xm/z” and there is a lack of information about the lipid composition and the m/z. Considering adding the info to the current supplemental table/figure will make the data more understandable.

Author Response

  1. In Figure 1, is the x-axis label “0,1” indicates “0.1”? “12 times decrease for PGR” in the result description is about 0.083. If the lowest value in x-axis is 0.1, 0.083 is out of range.

Answer: thank you for this valuable remark, y-axis at figure 1 was corrected.

  1. In supplemental table 1, all the lipids detected were indicated as “xm/z” and there is a lack of information about the lipid composition and the m/z. Considering adding the info to the current supplemental table/figure will make the data more understandable.

Answer: the mistake in table S1 was corrected – all data present m/z, not “xm/z”.